# Whole-organism 3D quantitative characterization of zebrafish melanin by silver deposition micro-CT

**Spencer R Katz[1,2,3]\*, Maksim A Yakovlev[1,2], Daniel J Vanselow[1,2], Yifu Ding[1,2,3], Alex Y Lin[1,2], Dilworth Y Parkinson[4], Yuxin Wang[5], Victor A Canfield[1,2], Khai C Ang[1,2,6], Keith C Cheng[1,2,6]\***

[1]Division of Experimental Pathology, Department of Pathology, Pennsylvania State University College of Medicine, Hershey, United States; [2]The Jake Gittlen Laboratories for Cancer Research, Penn State College of Medicine, Hershey, United States; [3]Medical Scientist Training Program, Penn State College of Medicine, Hershey, United States; [4]Advanced Light Source, Lawrence Berkeley National Labs, Berkeley, United States; [5]Mobile Imaging Innovations, Inc, Palatine, United States; [6]Zebrafish Functional Genomics Core, Penn State College of Medicine, Hershey, United States

**Abstract** We previously described X-ray histotomography, a high-resolution, non-destructive form of X-ray microtomography (micro-CT) imaging customized for three-dimensional (3D), digital histology, allowing quantitative, volumetric tissue and organismal phenotyping (Ding et al., 2019). Here, we have combined micro-CT with a novel application of ionic silver staining to characterize melanin distribution in whole zebrafish larvae. The resulting images enabled whole-body, computational analyses of regional melanin content and morphology. Normalized micro-CT reconstructions of silver-stained fish consistently reproduced pigment patterns seen by light microscopy, and further allowed direct quantitative comparisons of melanin content across wild-type and mutant samples, including subtle phenotypes not previously noticed. Silver staining of melanin for micro-CT provides proof-of-principle for whole-body, 3D computational phenomic analysis of a specific cell type at cellular resolution, with potential applications in other model organisms and melanocytic neoplasms. Advances such as this in whole-organism, high-resolution phenotyping provide superior context for studying the phenotypic effects of genetic, disease, and environmental variables.

\*For correspondence:
skatz2@pennstatehealth.psu.edu
(SRK);
kcheng76@gmail.com (KCC)

## Introduction

The zebrafish (*Danio rerio*) is an important vertebrate model organism for developmental biology, functional genomics, and large-scale drug and toxicological screens due to its high fecundity, transparent, externally developing embryos, relatively low husbandry costs, and small size (*Lieschke and Currie, 2007*). One of the most readily observable phenotypes in the zebrafish is its distinctive pigmentation patterning, which, in wild-type adults, appears as alternating dark and light longitudinal stripes. The study of the cellular components of these stripes, including brown-black melanophores, silvery iridophores, and yellow xanthophores, has taught us a lot about the necessary cells, genes, and pathways required for normal pigmentation development and the evolution of complex color patterning (*D'Agati et al., 2017*; *Frohnhöfer et al., 2013*; *Hultman and Johnson, 2010*; *Kelsh, 2004*; *Kelsh et al., 1996*; *Lister, 2002*; *Lister et al., 1999*; *Singh and Nüsslein-Volhard, 2015*). Disruption of melanogenesis or localization of melanophores, iridophores, or xanthophores has been found in a plethora of mutant strains with unique pigmentation patterns, or even virtually no body pigment, that are useful for screening or imaging studies (*Parichy, 2006*). For example, the *crystal* (*mitfa^{w2/w2}*;

$mpv17^{a9/a9}$; $slc45a2^{b4/b4}$) strain, a compound mutant with defects in both melanogenesis and iridophore development, never develops pigment and is used for applications where transparent post-embryonic organisms are required, such as fluorescence imaging (**Antinucci and Hindges, 2016**). Another pigmentation mutant, *golden* ($slc24a5^{b1/b1}$), is frequently used for mutagenesis screening or testing of gene editing technology because its single affected locus produces an early, dramatic reduction in eye and body melanin content that has been implicated in human skin color (**Doyon et al., 2008**; **Ginger et al., 2008**; **Hoshijima et al., 2016**; **Lamason et al., 2005**; **Moore et al., 2006**; **Streisinger et al., 1981**; **Streisinger et al., 1989**).

Melanin, the dark pigment produced in zebrafish melanophores, is a heterogeneous mix of cross-linked derivatives of the amino acid tyrosine. In humans, melanin is produced in skin melanocytes and transferred to keratinocytes which prevents UV-based DNA damage; this pigment is the major component of human skin and hair color with implications in human health and disease (**Slominski et al., 2004**). Dysfunction of human pigmentation is responsible for diseases such as albinism and vitiligo. Overgrowth of melanocytes can result in melanocytic nevi or malignant melanoma with darkly pigmented lesions. In addition, neuromelanin, produced as a byproduct of dopamine synthesis, stains the substantia nigra pars compacta in the basal ganglia, the region principally lost in Parkinson's disease (**Kumar et al., 2015**).

Despite the importance of melanin patterning and distribution for developmental and functional studies, assessments of pigmentation patterns and cell morphology have been largely descriptive due to the challenges of melanin quantification at cellular resolution, particularly in whole animals. In zebrafish, pigmentation studies have generally relied on light microscopy and two-dimensional photography for phenotyping, with pigmentation interpreted as counts of pigmented/unpigmented cells in a particular region of the fish in a manner that is, at best, semi-quantitative (**D'Agati et al., 2017**; **Frohnhöfer et al., 2013**; **Hultman and Johnson, 2010**; **Kelsh, 2004**; **Kelsh et al., 1996**; **Lister, 2002**; **Lister et al., 1999**; **Moore et al., 2006**; **Singh and Nüsslein-Volhard, 2015**). Quantitative measurement of melanin content in the three-dimensional (3D), in situ context of a whole sample poses a number of unique challenges due to the light-absorbing properties of the pigment. Fluorescence techniques may readily tag melanophores, but fluorescence is quenched or blocked by the melanin itself and such techniques are limited in penetration due to the light-scattering properties of soft tissues (**Antinucci and Hindges, 2016**; **Hultman and Johnson, 2010**). Electron microscopy techniques, while exquisitely detailed, provide only a small field of view (FOV) of pigment cells and lack 3D context (**Hirata et al., 2003**; **Hirata et al., 2005**). Photoacoustic/optoacoustic microscopy has been used to produce 3D pigmentation patterns in whole zebrafish larvae over time, but while such images can have high lateral resolution, they are limited in axial resolution by the ultrasound wavelength (**Kneipp et al., 2015**). Other imaging techniques have taken advantage of melanin's unique metal scavenging or binding properties, such as MRI or PET imaging, but these techniques do not provide the spatial resolution needed for detailed studies of pigment patterns (**Cheng et al., 2011**; **Enochs et al., 1997**; **Ren et al., 2009**).

X-ray microtomography (micro-CT) is a 3D imaging technique that uses a series of X-ray projections to reconstruct small samples at high, isotropic resolution without the need for sectioning. We previously reported an implementation of micro-CT optimized for cellular feature detection in whole zebrafish samples (histotomography) with soft tissue contrast enhancement through staining with phosphotungstic acid (**Ding et al., 2019**). We continue to develop staining procedures to selectively enhance particular microanatomical features to expand the potential of micro-CT imaging for screening and phenome projects (**Cheng et al., 2011**). Because reconstructed X-ray attenuation values are proportional to the amount of stained material in a sample, micro-CT images have been used for quantitative studies of development and gene expression, even in optically opaque samples (**Metscher and Müller, 2011**).

Here, we present a novel application of ionic silver staining for a micro-CT-based, 3D, quantitative assay for melanin in whole zebrafish samples. The procedure, based on the reduction of silver used for histological Fontana-Masson staining (**Bancroft, 2008**), appears highly specific to melanin and preserves native pigment distribution in whole specimens. To demonstrate the utility of this technique, we imaged wild-type zebrafish as well as mutant strains with unique pigmentation patterns, including *golden* and body pigment-deficient mutants (*nacre/casper*) (**D'Agati et al., 2017**; **Lister et al., 1999**; **White et al., 2008**), for visualization and quantitative analyses of zebrafish melanin.

# Materials and methods

**Key resources table**

| Reagent type (species) or resource | Designation | Source or reference | Identifiers | Additional information |
|---|---|---|---|---|
| Strain, strain background (*Danio rerio*) | Wild-type Tu | Zebrafish International Resource Center (ZIRC) | RRID:ZIRC_ZL57 | |
| Strain, strain background (*Danio rerio*) | Wild-type WIK | Zebrafish International Resource Center (ZIRC) | RRID:ZIRC_ZL84 | |
| Strain, strain background (*Danio rerio*) | *golden* (*slc24a5*$^{b1/b1}$) | *Lamason et al., 2005* | RRID:ZFIN_ZDB-GENO-071214-1 | |
| Strain, strain background (*Danio rerio*) | *nacre* (*mitfa*$^{w2/w2}$) | *Lister et al., 1999* | RRID:ZFIN_ZDB-GENO-990423-18 | |
| Strain, strain background (*Danio rerio*) | *casper* (*mitfa*$^{w2/w2}$; *mpv17*$^{a9/a9}$) | *D'Agati et al., 2017*; *White et al., 2008* | RRID:ZFIN_ZDB-GENO-160210-11 | |
| Strain, strain background (*Danio rerio*) | *crystal* (*mitfa*$^{w2/w2}$; *mpv17*$^{a9/a9}$; *slc45a2*$^{b4/b4}$) | *Antinucci and Hindges, 2016* | RRID:ZFIN_ZDB-GENO-160927-1 | |
| Commercial assay or kit | Fontana-Masson Staining Kit | Sigma-Aldrich | Cat#:HT200 | Included gold chloride not used |
| Chemical compound, drug | Tricaine-S | Syndel USA | Cat#:MS-222 | |
| Chemical compound, drug | 10 % Neutral buffered formalin | Fisher Scientific | Cat#:SF100 | |
| Chemical compound, drug | LR White resin, catalyzed | Electron Microscopy Sciences | Cat#:14,383 | |
| Software, algorithm | TomoPy | *Gürsoy et al., 2014* | RRID:SCR_021359 | |
| Software, algorithm | Gridrec | *Dowd et al., 1999*; *Rivers, 2012* | RRID:SCR_021358 | |
| Software, algorithm | Fiji/ImageJ2 | *Schindelin et al., 2012* | RRID:SCR_002285 | |
| Software, algorithm | Avizo 3D Software | Thermo Fisher Scientific | RRID:SCR_014431 | Versions 2020.1, 2020.2 |
| Software, algorithm | IBM SPSS Statistics | IBM | RRID:SCR_019096 | Version 27.0.1.0 |
| Software, algorithm | Three.js | https://threejs.org/ | RRID:SCR_021357 | For online viewer at http://3d.fish/stains/silver/ |
| Other | Polyimide tubing | Small Parts, Inc | | Inner diameter 0.0403" |

## Animals

Zebrafish were housed in a recirculating system at an average temperature of 28 °C on a 14 hr:10 hr light:dark cycle. Fish were fed three times a day a diet consisting of brine shrimp and flake food. Embryos and larvae were raised in darkness and staged according to a standard developmental staging series (*Kimmel et al., 1995*). Wild-type (a mixed Tu/WIK background) and *golden* (*slc24a5*$^{b1/b1}$) zebrafish (*Lamason et al., 2005*) were bred and maintained as homozygotes. Other pigmentation variants were produced as follows: Compound *mitfa*$^{w2/w2}$; *mpv17*$^{a9/+}$; *slc45a2*$^{b4/+}$ mutants (phenotypically *nacre*) were in-crossed to produce offspring that were separated by eye melanin phenotype prior to fixation/staining. *Mitfa*$^{w2/w2}$ (*nacre*) and *mitfa*$^{w2/w2}$; *mpv17*$^{a9/a9}$ (*casper*) offspring possessed pigmented eyes while *mitfa*$^{w2/w2}$; *slc45a2*$^{b4/b4}$ and *mitfa*$^{w2/w2}$; *mpv17*$^{a9/a9}$; *slc45a2*$^{b4/b4}$ (*crystal*) offspring lacked eye pigment; representatives from the pigmented eye population (mixed *nacre* and *casper* larvae) were analyzed for this study and are referred to by the *mitfa*$^{w2/w2}$ background.

## Sample preparation and silver staining

Wild-type and mutant zebrafish were each subjected to the same fixation protocol, as described previously (*Lin et al., 2018*). Briefly, 5 days post-fertilization (dpf) larvae were euthanized in pre-chilled Tricaine-S (MS-222, 200 mg/L) solution (Syndel USA, Ferndale, WA) buffered with 1 % phosphate buffered saline (PBS) for at least 60 s and fixed overnight (>20 hr) in chilled 10 % neutral buffered formalin (Fisher Scientific, Allentown, PA) in a flat-bottom glass container at room temperature with gentle

agitation. Fixed specimens were washed three times in cold 1 × PBS for 5 min each and stored in 1 × PBS at 4 °C until proceeding with staining.

All samples were stained at the same time under the same conditions using a modified Fontana-Masson Kit staining protocol (Sigma-Aldrich Procedure #HT200). Prior to staining, ammoniacal silver solution was produced in a chemical fume hood by adding concentrated ammonium hydroxide solution dropwise to a 2.5 % silver nitrate solution until the resulting tan-brown precipitate dissolved, then left undisturbed overnight until the solution stayed clear. Also prior to staining, each individual fixed larva was mounted in 3 % methylcellulose on a glass slide for photography of the top (35 × and 100 × zoom) and side (35 ×) of each sample on a ZEISS Axio Zoom.V16 microscope (Carl Zeiss Microscopy, LLC, White Plains, NY) for comparison with X-ray images. Individual larvae were washed three times for 5 min each in ddH$_2$O and then stained for 18 hr in ammoniacal silver solution in flat-bottom glass containers with gentle agitation at room temperature. As a control, some larvae were incubated instead with ddH$_2$O alone. Although the histological Fontana-Masson staining procedure typically includes a gold toning step (whereby a gold chloride wash replaces silver with gold), pilot experiments indicated that this step did not increase X-ray contrast and it was thus eliminated in subsequent procedures (data not shown). Following staining, larvae were washed three times with ddH$_2$O and post-fixed in 5 % sodium thiosulfate solution for 10 min, followed by a final three washes with ddH$_2$O to prepare samples for embedding.

## Micro-CT imaging

Samples were embedded for micro-CT imaging in a manner described previously (*Lin et al., 2018*). Briefly, stained samples were dehydrated through an alcohol series of 35 % (20 min), 50 % (20 min), 70 % (30 min), 90 % (30 min), 95 % (30 min), and 100 % (30 min, two times) ethanol solutions, and submerged overnight in a 1:1 solution of ethanol and LR White acrylic resin (Electron Microscopy Sciences, Hatfield, PA) with gentle agitation at room temperature. Samples were submerged in 100 % LR White resin for 2 hr followed by fresh resin for 1 hr with gentle agitation at room temperature, and then embedded in polyimide tubing (Small Parts, Inc, Miami Lakes, FL; inner diameter 0.0403") for 24 hr at 65 °C.

Micro-CT imaging was performed on Beamline 8.3.2, Tomography (micro-CT), at the Advanced Light Source (ALS) at the Lawrence Berkeley National Laboratory (LBNL, Berkeley, CA). A double-multilayer monochromator was used to select an X-ray energy of 26 keV to optimize silver-based contrast (elemental silver K edge = 25.5 keV) (*Hubbell and Seltzer, 2004*). Acquisition was performed using a custom detector system based on a 6 × objective lens with 0.6 numerical aperture and a 101 megapixel thermo-electrically cooled CMOS camera outputting a 11,648-pixel × 8742-pixel image (Vision Systems Technology, Vista, CA). Horizontal FOV was approximately 5 mm, but projections were cropped to cover the width of the sample tube as they were acquired. Whole 5 dpf larval zebrafish were imaged over two scans, covering head and tail regions, respectively. Each constant motion scan resulted in 1017 projections over 180° with an exposure time of 175 ms per projection. A sample to scintillator distance of 33 mm was chosen to optimize phase effect. Two flat field images were acquired per sample, one before the head segment and one after the tail segment, to be used for image normalization.

## Reconstruction, post-processing, and analysis

Flat field correction, ring artifact reduction, and image reconstruction were performed using the open source TomoPy toolkit (*Gürsoy et al., 2014*). Flat field correction was performed using the flat field image taken either before or after sample scanning, depending on which produced the best contrast on test projections. Images were reconstructed using Gridrec (*Dowd et al., 1999*; *Rivers, 2012*) with ring artifact reduction (*Miqueles et al., 2014*; *Münch et al., 2009*) and a second-order Butterworth filter with a cutoff of 0.2 to reduce noise, resulting in a nominal isotropic voxel size of 0.52 µm, as estimated by the number of reconstructed voxels spanning the 1.03 mm outer diameter of the reconstructed polyimide sample tube.

Thirty-two-bit reconstructed images were further processed using Fiji (*Schindelin et al., 2012*). Reconstructions were cropped and rotated, and regions containing the zebrafish sample were segmented on a per slice basis to remove the plastic tube and any bubbles which had formed near the sample. This 'cleaned up' data was normalized to the LR White resin present in every sample to

allow direct comparisons between scans as follows: An average attenuation coefficient for the resin in each reconstruction ($\mu_{resin}$) was determined as the mean intensity value over all slices of a 100-pixel × 100-pixel square of empty resin, distant from the sample or sides of the tube. Subsequently, each reconstruction was processed using the formula

$$\mu_{normalized} = \frac{\mu - \mu_{resin}}{\mu_{resin}}$$

where $\mu$ is the attenuation value at any given pixel in a reconstruction and $\mu_{normalized}$ is the normalized intensity. Normalized reconstructions were converted to 16-bit for visualization and analysis using the same minimum (–11) and maximum (155) $\mu_{normalized}$ for all reconstructions.

3D rendering and analysis, including registration of head and tail segments, were performed using Avizo versions 2020.1 and 2020.2 (Thermo Fisher Scientific). A combination of thresholding and manual segmentation was used to assign voxels in merged datasets to established larval pigmented regions, including left and right retinal pigment epithelia (RPE), dorsal stripe, ventral stripe, yolk sac stripe, and left and right lateral stripes. Regions of staining intensity not associated with one of these regions were assigned to an 'other pigment' category. To clean up the segmentations by closing small holes and reducing noise, the morphological operator 3D Closing (size = 5 px), followed by 3D Opening by Reconstruction (size = 1 px), was used on the label file for each dataset. Intensity statistics and volume information for the pigmented regions described above were then extracted from the resulting segmentations. Statistical analysis was performed using IBM SPSS Statistics 27.0.1.0 (IBM).

## Results and discussion
### Ionic silver stains reproduce 3D pigmentation patterns in micro-CT images

To explore a 3D, quantitative assay for melanin, we developed a sample preparation and imaging strategy for micro-CT, an X-ray-based imaging technique, using silver. Activated silver ion solutions, such as Fontana-Masson, have been used for staining melanin in histology slides because the anionic melanin reduces the silver cations to solid silver, which appears dark on light microscopy (*Bancroft, 2008*) and attenuates X-rays. Because a silver deposition strategy has also been used for molecular imaging in micro-CT of chick embryos (*Metscher and Müller, 2011*), we reasoned that ionic silver staining could be adapted for in situ imaging of zebrafish pigmentation.

A general overview of our staining strategy is shown in *Figure 1A*. Silver nitrate was activated with ammonium hydroxide and the resulting ammoniacal silver solution was reacted with endogenous melanin to deposit solid silver; silver deposition was assumed to be proportional to the local concentration of melanin in a given sample. Staining times were increased from 30 min to 18 hr for whole-mount samples, as opposed to histology slides, to account for the diffusion time necessary to perfuse thick tissue; longer incubation times produced greater CT contrast with minimal background staining. Some non-specific silver deposition blackened the outside of stained samples, but this concentration of silver was not sufficient to be visible on micro-CT images. After sodium thiosulfate post-fixation to remove unreacted silver, samples were embedded in resin and imaged at the Tomography beamline 8.3.2 of the ALS synchrotron at LBNL with a beam energy optimized for silver at 26 keV. Elemental silver (K edge = 25.5 keV) (*Hubbell and Seltzer, 2004*) in the stained samples attenuated X-rays to provide contrast on the resulting micro-CT reconstructions. Under the same conditions, an unstained wild-type zebrafish exhibited negligible X-ray attenuation and no pigment pattern upon reconstruction (*Figure 1—figure supplement 1*).

We chose to scan 5 dpf larvae because, at this stage, the majority of wild-type larval pigment has developed, including the RPE and body melanophores (*Hultman and Johnson, 2010*). A single slice from a representative reconstruction of a 5 dpf wild-type larva that was rotated and cropped to the region of interest is presented in *Figure 1B*. Several key features of the reconstruction have been highlighted, including the high intensity regions of stained sample (RPE and body pigment), the resin embedding medium, and the inner and outer walls of the polyimide tubing used for embedding. By dividing the number of voxels spanning the polyimide tube by its outer diameter (1.03 mm), we estimated our reconstructed isotropic voxel resolution at 0.52 μm; isotropic resolution enabled rotation and re-slicing of the volume without distortion. We also noted that microscopic air bubbles formed in all samples at the tube-resin interface, sometimes close to the zebrafish sample, due to resin

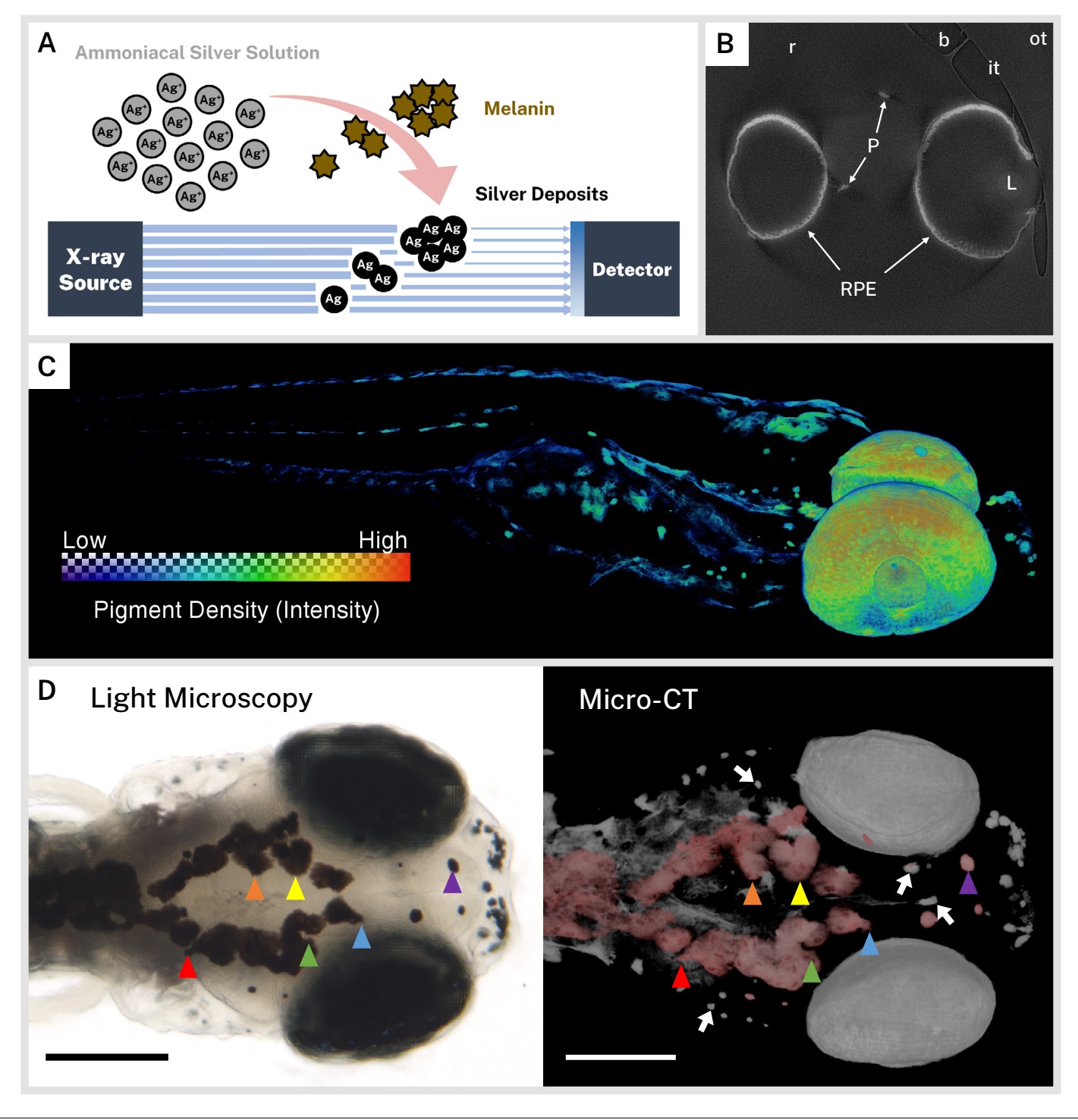

**Figure 1.** Ionic silver stains reproduce three-dimensional (3D) pigmentation patterns in X-ray microtomography (micro-CT) images. (**A**) Schematic overview of staining procedure. Ammoniacal silver solution reacts with endogenous melanin to deposit solid silver, which attenuates X-rays in micro-CT imaging. (**B**) A representative cropped and rotated slice of a micro-CT reconstruction of a 5 days post-fertilization (dpf) wild-type zebrafish stained with silver. RPE = retinal pigment epithelium, P = body pigment, L = lens, r = resin, it = inner wall of sample tube, ot = outer wall of sample tube, b = air bubble. (**C**) 3D rendering of a 5 dpf wild-type zebrafish stained with silver with a heatmap to illustrate pigment density throughout the fish. (**D**) Corresponding light microscopy (left) and micro-CT (right) images exhibit the same pigmentation patterns. In the micro-CT image, a top-down 3D rendering of stained melanin is shown in grayscale with the dorsal-most melanin digitally colored red to aid comparison with the light micrograph. Some distinguishing shared features are highlighted with colored arrowheads. White arrows indicate deeper melanin obscured by soft tissue in the light

*Figure 1 continued on next page*

*Figure 1 continued*

micrograph that can be visualized by targeted 3D re-rendering. Scale bars = 200 µm. Unstained samples do not exhibit melanin-related attenuation (*Figure 1—figure supplement 1*).

The online version of this article includes the following figure supplement(s) for figure 1:

**Figure supplement 1.** Unstained zebrafish larvae do not exhibit pigment pattern intensity in X-ray microtomography (micro-CT) reconstructions.

shrinkage during polymerization. We manually segmented and removed bubbles and the sample tube from reconstructions to eliminate their interference with visualization and analysis.

To cover the body length of the larval zebrafish within the synchrotron X-ray beam, we imaged each fish over two overlapping FOVs by vertically translating the sample between acquisitions. Each of the two resulting reconstructions were later registered and combined for subsequent visualization and analysis. A 3D rendering of a representative volume (*Figure 1C*) with a color heatmap suggests that ionic silver staining produces a 3D pigmentation map throughout the whole larvae with minimal background. To confirm the ability of the micro-CT images to faithfully reproduce larval pigmentation patterns, we compared 3D renderings of reconstructed volumes with light micrographs of the same fish taken prior to silver staining (*Figure 1D*, *Video 1*). The top of the head represents a unique 'fingerprint' pattern for each larva, and all the surface melanin features evident in the light micrograph were also observed in the micro-CT image. Pigment units lying deep in the fish that are obscured by soft tissue in light microscopy were readily discernable in micro-CT reconstructions, demonstrating the sensitivity of this new assay.

## Characterizing wild-type larval zebrafish melanin pigmentation in 3D

The wild-type larval zebrafish forms a stereotypical ontogenetic pigmentation pattern with pigment cell precursors starting from the neural crest at the dorsum and spreading ventrally, eventually forming four longitudinal 'stripes' of melanin by approximately 3 dpf: dorsal, lateral, ventral (above the air bladder), and yolk sac (below the yolk) (*Kelsh, 2004*; *Kelsh et al., 1996*). After this pattern is formed from direct-developing melanophores, pigment stem cells fill in gaps in the patterns or are responsible for pigment cell regeneration such that the majority of the larval pigment pattern has matured by

5 dpf (*Hultman and Johnson, 2010*). The longitudinal stripes of larval pigment were readily identifiable in our wild-type larvae (*Figure 2A*). To better examine the distribution of pigment throughout the fish, we used a combination of thresholding and segmentation on our micro-CT reconstructions to assign pigment to anatomical regions in our larvae. In 3D, we observed the dorsal, ventral, and yolk sac stripes as flattened sheets of pigment cells that often met in the midline of the fish and followed the curves of body structures (*Figure 2B–C*, *Video 1*). In contrast, the two lateral stripes were made up of intermittent thin, flat accumulations of melanin, one stripe on each side of the larva (*Figure 2B–C*).

Zebrafish can adapt their pigmentation to their environment for camouflage; larval zebrafish raised on white backgrounds aggregate melanin within melanophores while those raised on black backgrounds disperse melanin throughout the melanophores (*Logan et al., 2006*). Our larvae, reared in a closed, dark incubator, exhibited a dispersed staining phenotype on micro-CT with sheets of pigment and indistinct cell boundaries. Notably, the dorsal, ventral, and yolk sac stripes exhibited ovoid transparencies in the melanin

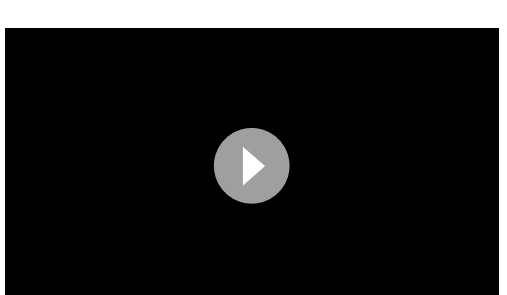

**Video 1.** Wild-type zebrafish melanin. This video shows an overview of melanin staining with silver by X-ray microtomography (micro-CT) imaging in a representative wild-type 5 days post-fertilization (dpf) zebrafish larva. The three-dimensional (3D) volume is generated from a series of 0.52 µm slices and reproduces pigmentation patterns seen by light microscopy (colored arrowheads). In a view of the top layer of the dorsal melanin stripe (false colored for clarity), ovoid transparencies are observed representing large organelles lacking melanin (circles). An isolated volume of the retinal pigment epithelia (RPE) exhibits the fused choroidal fissure and the optic nerve egress. Finally, a 360° view of the larval melanin, colored by anatomic region, is shown. Background music by DJ Vanselow.

https://elifesciences.org/articles/68920/figures#video1

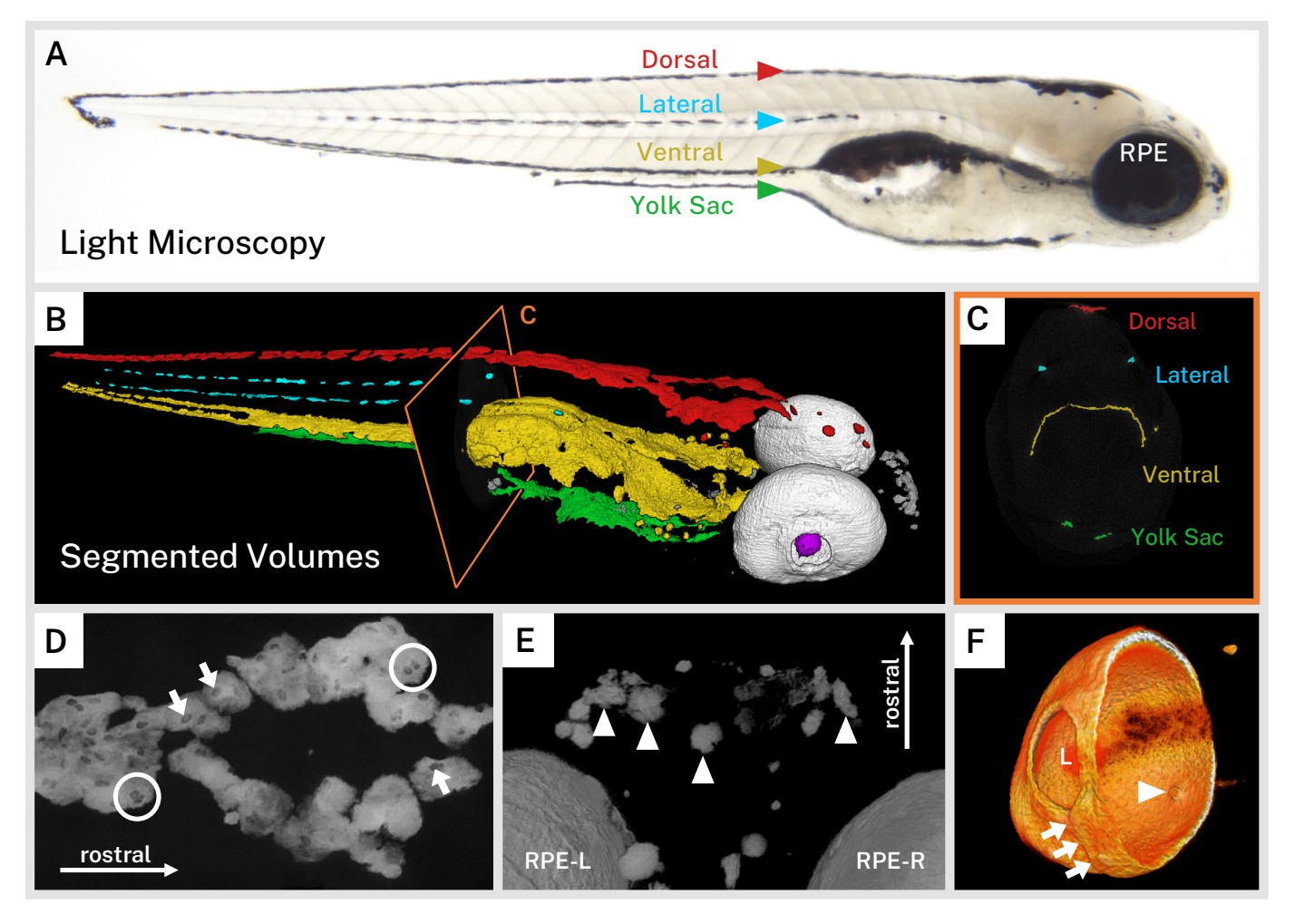

**Figure 2.** Silver-based X-ray microtomography (micro-CT) enables segmentation and visualization of melanin microanatomy in three dimensions (3D). (**A**) A light micrograph of the side of a 5 days post-fertilization (dpf) wild-type larva indicating the major regions of melanin pigment: dorsal, lateral, ventral, and yolk sac stripes and the retinal pigment epithelium (RPE). (**B**) A 3D rendering with orthoslice (**C**) of micro-CT volumes segmented into anatomical regions shows the organization of larval pigment into layers. Red = dorsal stripe, yellow = ventral stripe, green = yolk sac stripe, cyan = lateral stripes, white = RPE, gray = other body melanin, purple = lens. (**C**) Single slice of micro-CT data with color overlay corresponding to the indicated region in B. (**D**) 200-slice maximum intensity projection of dorsal stripe melanin exhibits transparencies in the staining indicating the position of large organelles (arrows) including potentially binucleated cells (circles). A sample of these transparencies was measured to estimate average size (*Figure 2— figure supplement 1*, *Figure 2—source data 1*). (**E**) View from the top-down of a volume rendering showing rostral melanin in the nose forming globular, dendritic cells (arrowheads). RPE-L = left RPE, RPE-R = right RPE. (**F**) Isolated volume rendering of the right eye with a clipping plane showing the villous inner surface and smooth outer surface of the RPE. The rendering has been falsely colored by intensity to highlight certain anatomical features, including local pigment thickness variability throughout the RPE, the egress of the optic nerve (arrowhead), and the fused choroidal fissure (arrows). L = lens.

The online version of this article includes the following source data and figure supplement(s) for figure 2:

**Source data 1.** Measurements of ovoid transparencies in melanin staining.

**Figure supplement 1.** Measurements of ovoid transparencies in melanin staining.

staining that are approximately 9.29 μm in length (SD = ±1.23 μm, *Figure 2D*, *Figure 2—figure supplement 1*, and *Video 1*). We suspect that these transparencies represent large organelles, likely nuclei, which are devoid of melanin. This observation is consistent with high-resolution photographs of black background-adapted dorsal melanophores reported previously (*Logan et al., 2006*). Several of these transparencies were in close proximity (*Figure 2D*); zebrafish melanophores have been frequently shown to be multinucleated (*Usui et al., 2018*). In contrast to the flattened, diffuse melanin

in the stripes, pigment units not associated with the four larval stripes tended to have a globular, condensed morphology. These cells, deeper in the soft tissue of the fish, were individually separated and dendritic, particularly in the anterior portion of the head (*Figure 2E*).

Another major structure visible in our silver-stained wild-type samples was the eye due to the melanin in the RPE. The RPE is one of the outermost layers of the retina, contacting the choroidal vasculature on its outer surface and the outer nuclear layer of the retina on its inner surface. It is a critical structure for vertebrate visual function. Unlike the neural crest-derived body melanophores, the RPE develops from the optic neuroepithelium and is closely associated with the photoreceptor cells of the developing retina (*Antinucci and Hindges, 2016*; *Bharti et al., 2006*; *Gestri et al., 2012*; *Schmitt and Dowling, 1999*). With its dense concentration of melanin, the RPE stained brightly in our wild-type samples. Digitally cutting through the zebrafish eye revealed the major morphological features of this layer, which has local thickness variability (*Figure 2F*, *Video 1*). The outer surface was relatively smooth and contiguous, except for a ridge where the choroidal fissure fused during development. The inner surface exhibited regular villous protrusions toward the center of the eye where the apical side of the RPE cells invaginate into the inner nuclear layer. A hole in the ventromedial portion of the RPE indicated the location of the optic nerve egress. Surrounding the back of the eye, it was also possible to see portions of the sclera, which has its own pigment derived from body melanophores (*Gestri et al., 2012*). Interestingly, despite not having melanin, the dense, fibrous lens stained weakly with silver and its center was partially visible in our reconstructions, providing consistent anatomical landmarks across all samples.

One of the advantages of micro-CT reconstructions is that they are inherently quantitative, with information at each voxel about the local concentration of stained material. Having established a

**Table 1.** Quantification of volume and cumulative sum of intensity for wild-type samples for segmented pigment regions.

| Wild-type specimen | #1 | #2 | #3 | Average | SD |
|---|---|---|---|---|---|
| **Pigment region** | **Segmented volume ($\mu m^3$)** | | | | |
| Total | $4.81 \times 10^6$ | $4.52 \times 10^6$ | $5.27 \times 10^6$ | $4.87 \times 10^6$ | $\pm 3.78 \times 10^5$ |
| RPE (right) | $1.20 \times 10^6$ | $1.02 \times 10^6$ | $1.28 \times 10^6$ | $1.17 \times 10^6$ | $\pm 1.31 \times 10^5$ |
| RPE (left) | $1.25 \times 10^6$ | $9.65 \times 10^5$ | $1.46 \times 10^6$ | $1.23 \times 10^6$ | $\pm 2.49 \times 10^5$ |
| Dorsal stripe | $5.45 \times 10^5$ | $4.87 \times 10^5$ | $5.11 \times 10^5$ | $5.15 \times 10^5$ | $\pm 2.93 \times 10^4$ |
| Ventral stripe | $1.41 \times 10^6$ | $1.56 \times 10^6$ | $1.62 \times 10^6$ | $1.53 \times 10^6$ | $\pm 1.10 \times 10^5$ |
| Yolk sac stripe | $2.29 \times 10^5$ | $3.63 \times 10^5$ | $2.06 \times 10^5$ | $2.66 \times 10^5$ | $\pm 8.52 \times 10^4$ |
| Lateral stripe (right) | $5.24 \times 10^4$ | $4.10 \times 10^4$ | $6.73 \times 10^4$ | $5.36 \times 10^4$ | $\pm 1.32 \times 10^4$ |
| Lateral stripe (left) | $5.91 \times 10^4$ | $2.96 \times 10^4$ | $6.16 \times 10^4$ | $5.01 \times 10^4$ | $\pm 1.78 \times 10^4$ |
| Other | $6.72 \times 10^4$ | $5.89 \times 10^4$ | $6.79 \times 10^4$ | $6.47 \times 10^4$ | $\pm 5.02 \times 10^3$ |
| **Pigment region** | **Cumulative sum of intensity (normalized units)** | | | | |
| Total | $5.05 \times 10^{11}$ | $4.71 \times 10^{11}$ | $4.73 \times 10^{11}$ | $4.83 \times 10^{11}$ | $\pm 1.94 \times 10^{10}$ |
| RPE (right) | $1.48 \times 10^{11}$ | $1.43 \times 10^{11}$ | $1.40 \times 10^{11}$ | $1.44 \times 10^{11}$ | $\pm 4.45 \times 10^9$ |
| RPE (left) | $1.74 \times 10^{11}$ | $1.47 \times 10^{11}$ | $1.57 \times 10^{11}$ | $1.59 \times 10^{11}$ | $\pm 1.39 \times 10^{10}$ |
| Dorsal stripe | $4.88 \times 10^{10}$ | $4.02 \times 10^{10}$ | $4.01 \times 10^{10}$ | $4.30 \times 10^{10}$ | $\pm 5.00 \times 10^9$ |
| Ventral stripe | $1.01 \times 10^{11}$ | $1.06 \times 10^{11}$ | $1.06 \times 10^{11}$ | $1.05 \times 10^{11}$ | $\pm 3.20 \times 10^9$ |
| Yolk sac stripe | $1.67 \times 10^{10}$ | $2.33 \times 10^{10}$ | $1.42 \times 10^{10}$ | $1.81 \times 10^{10}$ | $\pm 4.66 \times 10^9$ |
| Lateral stripe (right) | $5.30 \times 10^9$ | $3.64 \times 10^9$ | $5.52 \times 10^9$ | $4.82 \times 10^9$ | $\pm 1.03 \times 10^9$ |
| Lateral stripe (left) | $5.14 \times 10^9$ | $2.33 \times 10^9$ | $4.58 \times 10^9$ | $4.02 \times 10^9$ | $\pm 1.49 \times 10^9$ |
| Other | $5.73 \times 10^9$ | $4.91 \times 10^9$ | $5.21 \times 10^9$ | $5.28 \times 10^9$ | $\pm 4.17 \times 10^8$ |

Raw data provided in Table 1—source data 1.

The online version of this article includes the following source data for table 1:

**Source data 1.** Source data for quantification of volume and cumulative sum of intensity for wild-type samples for segmented pigment regions.

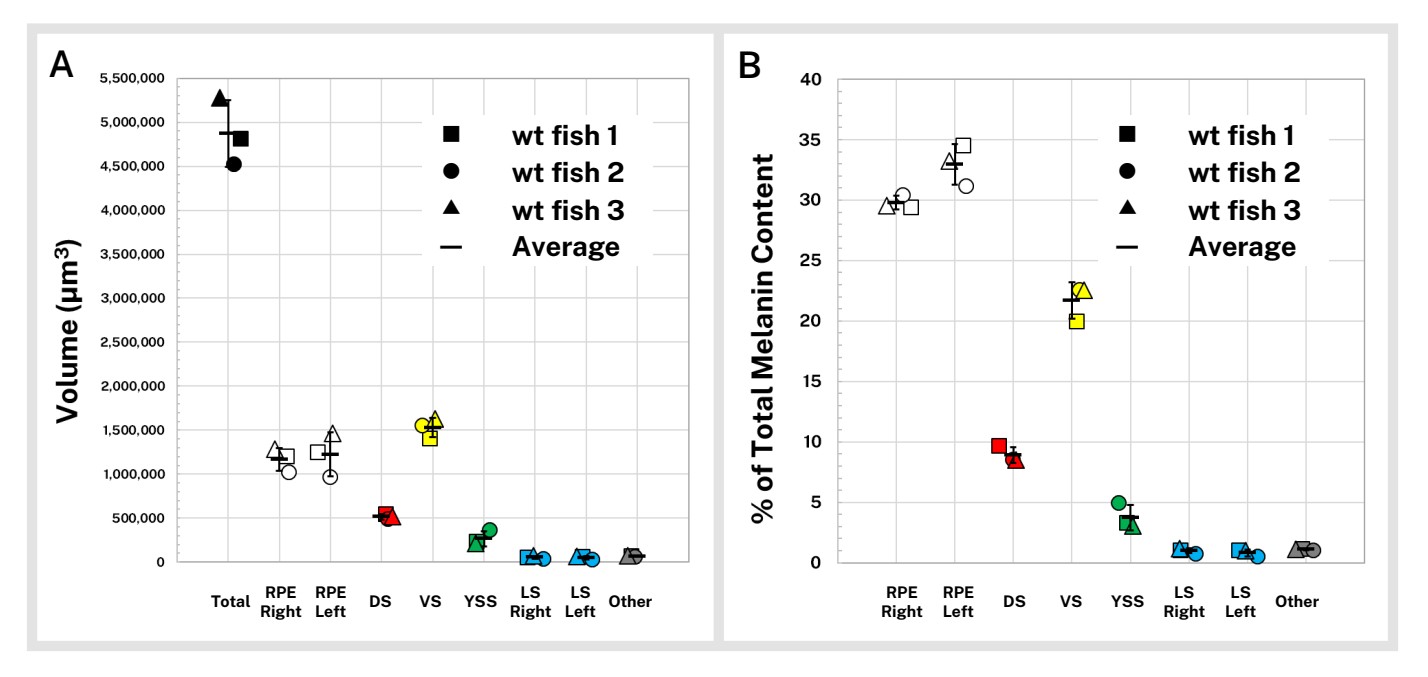

**Figure 3.** Quantification of silver-stained, X-ray microtomography (micro-CT)-scanned, and segmented larvae reveals wild-type melanin volume and content trends. Wild-type (wt) 5 days post-fertilization (dpf) larvae (n = 3) were stained with silver and micro-CT imaged under the same conditions then segmented into major pigment regions as described in *Figure 2*: retinal pigment epithelium (RPE, right and left), dorsal stripe (DS), ventral stripe (VS), yolk sac stripe (YSS), lateral stripes (LS, right and left), and other melanin (other). (**A**) Volumes of the total pigmented regions and each segmented region are shown for the three fish. (**B**) Reconstructed stain intensity values are assumed to be proportional to melanin density; integrated intensity values for the segmented regions represents the melanin content of these regions. As proportions of total melanin content (summed across all segmented regions) the wt fish show high concordance between individual samples. Error bars = standard deviation from average.

qualitative visualization of wild-type larval melanin patterning, we sought quantitative characterization of melanin content in the anatomical regions described above. The wild-type 5 dpf larval siblings (n = 3) used in this study were stained and imaged under the same conditions, and reconstructed intensity values were normalized to the average intensity of the embedding resin for all of our samples. The resulting normalized intensity values, although arbitrary units, were assumed proportional to the local density of silver deposition, and therefore melanin concentration, throughout each sample and could be compared directly between individual stained fish.

We first compared the volumes of the pigmented regions of the wild-type fish, including the right and left RPE, dorsal, ventral, yolk sac, right and left lateral stripes, and other body melanin not attributable to the four main stripes (*Table 1*, *Figure 3A*). Total melanin-containing volume covered $4.87 \times 10^6 \, \mu m^3$ (SD = $\pm 3.78 \times 10^5 \, \mu m^3$) of the average 5 dpf wild-type larva, much of which was distributed between the two eyes. The melanin-containing volumes of symmetrical structures (the RPE and lateral stripes) for each fish were similar, and standard deviation of the segmented regions, as percentage of the average melanin volumes between siblings, was on par with individual volume variation between brain regions of 5 dpf zebrafish (±11.9%) reported previously (*Ding et al., 2019*).

Although volume segmentation involves thresholding and intensity-based selection of stained regions of the micro-CT reconstruction, melanin volume measurements are independent of intensity and as such do not inform about the actual melanin content of the samples, which is not distributed evenly throughout the fish. To characterize melanin content for the wild-type larvae, we calculated intensity statistics for our segmented regions. Reported in normalized units, the cumulative sum of the intensity of each voxel in a volume is proportional to the total concentration of melanin in the volume and can be used to compare melanin content between normalized samples and within anatomical regions (*Table 1*). In *Figure 3B* we report the melanin content of the wild-type larvae as percentages of total melanin; 62.8 % (SD = ±1.1%) of the melanin in the average 5 dpf wild-type larva was contained in the RPE. Of the remaining body melanin, most of this pigment was contained within

the ventral stripe, followed by the dorsal, yolk sac, and lateral stripes. Approximately 1.1 % (SD = ±0.05%) of stained material was not associated with the eyes or four main stripes, mostly in the nose (*Figure 2E*). Distribution of melanin was highly concordant between the three scanned larvae.

## Comparing wild-type and mutant pigmentation variants by silver staining micro-CT

Dozens of zebrafish lines with mutations affecting larval and adult pigmentation patterns have been identified. To test the utility of our silver staining procedure with micro-CT imaging to compare aberrant melanin pigmentation to wild-type patterns, we stained representative fish from several mutant lines exhibiting larval pigmentation phenotypes that are commonly used in zebrafish genetic screens and imaging studies. Because all samples were normalized to the same background material, the mutant fish were compared directly to wild-type fish using the same visualization settings and computational analyses.

The *golden* zebrafish line is a prototypical model of human skin color, with a mutation in the potassium-dependent sodium/calcium exchanger gene *slc24a5* (*slc24a5$^{b1/b1}$*). A mutation in the human homolog of this gene, *SLC24A5*, is a major contributor to the skin color differences between people of African and European ancestry. The *slc24a5$^{b1/b1}$* zebrafish line has a delay in larval pigmentation and a dramatic reduction in larval and adult melanin content throughout the body and eyes. At the electron microscope level, *golden* melanophores contain fewer melanosomes that are less densely pigmented and irregularly shaped (*Lamason et al., 2005*). This early pigment phenotype also makes the *golden* mutant popular for studies such as screens or genetic engineering (*Doyon et al., 2008*; *Hoshijima et al., 2016*; *Moore et al., 2006*). As opposed to wild-type reconstructions, which exhibited intense whole-body staining on micro-CT imaging (*Figure 4A*), the most striking characteristic of the silver-stained *slc24a5$^{b1/b1}$* larvae was a globally diminished intensity corresponding to the decreased melanin content in these mutants (*Figure 4D*, *Video 2*). In addition, while wild-type pigment reconstructions formed essentially contiguous sheets down the length of the larval body (*Figure 4A–B*), melanin staining in the *golden* mutant was discontinuous and fragmented throughout (*Figure 4D–E*). To compare the overall pigmentation patterns of the larvae, we segmented our mutant fish into the same anatomical regions as we did for the wild-type fish (*Figures 2B and 4B*). Although melanin was diminished throughout the body, the dorsal, lateral, ventral, and yolk sac stripes, as well as other melanin deposits in the nose and eyes, were all identified in the *golden* mutant in an organization similar to the pigmentation in wild-type larvae (*Figure 4E*, *Video 2*). Single slices of the micro-CT reconstructions through the middle of the eyes of wild-type (*Figure 4C*) and *golden* (*Figure 4F*) larvae indicated that in addition to less intense staining, the *golden* fish had thinner stained areas in the RPE.

We also sought to examine a zebrafish mutant that lacks body pigment since these lines are commonly used in imaging studies when a transparent sample is desired. The *mitfa$^{w2/w2}$* mutation results in the *nacre* zebrafish line, which lacks all body melanin but retains the optic neuroepithelium-derived RPE melanophores (*Lister et al., 1999*). An additional mutation in iridophore development gene *mpv17* (*mpv17$^{a9/a9}$*) produces the compound mutant line *casper*, notable for its transparent adult body useful for fluorescence imaging (*D'Agati et al., 2017*; *White et al., 2008*). To examine these mutants, we bred *mitfa$^{w2/w2}$*; *mpv17$^{a9/+}$* larvae to produce a mix of both *nacre* and *casper* offspring, which are indistinguishable from each other in terms of melanin pigmentation at 5 dpf. Although it is possible to distinguish them using iridophore presence (*nacre*) or absence (*casper*) (*White et al., 2008*), larvae used in this study were only screened based on melanin and thus are referred to as *mitfa$^{w2/w2}$* background or *nacre*/*casper* fish throughout. As expected, larvae with the *mitfa$^{w2/w2}$* mutant background exhibited staining almost entirely in the RPE in our micro-CT scans, and as such we only used the head segment for visualization and analysis (*Figure 4G*, *Video 2*). Staining intensity, size, and shape of the eyes were similar to those of wild-type larvae, but body-stripe pigment or melanin deposits in the nose were completely absent. We segmented the *mitfa$^{w2/w2}$* larvae into component volumes consisting of the left and right RPE and any other stained material over a certain background intensity. A rendering of these segmented volumes (*Figure 4H*, *Video 2*) confirmed the absence of the dorsal, lateral, ventral, and yolk sac stripes, and also highlighted disorganized argentaffin material posterior to the eyes. Such material was also present in the wild-type and *golden* samples (included in the 'other' staining category), but was generally overshadowed by the intensely staining melanin. Single micro-CT slices through the eyes of the *mitfa$^{w2/w2}$* mutant revealed an RPE layer of similar

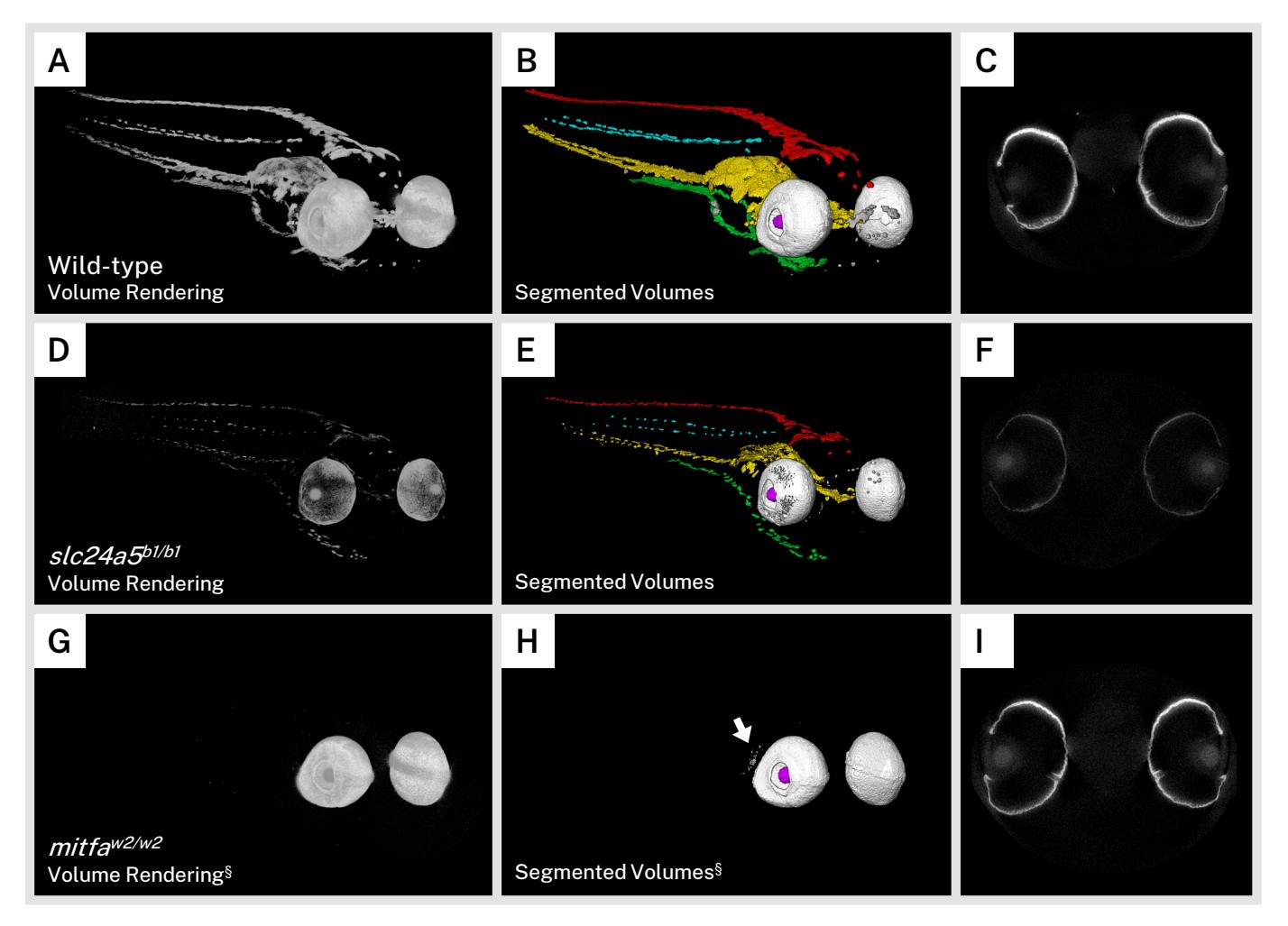

**Figure 4.** Silver staining and X-ray microtomography (micro-CT) of mutant zebrafish enables comparison of melanin content and organization with wild-type larvae. Volume renderings (**A, D, G**), segmented volumes (**B, E, H**), and representative single slices of micro-CT reconstructions through the retinal pigment epithelium (RPE; **C, F, I**) of representative 5 days post-fertilization (dpf) wild-type (**A–C**), *slc24a5*[b1/b1] (*golden*; **D–E**), and *mitfa*[w2/w2] (*nacre* or *casper*; **G–I**) larvae. In the segmented volumes, red = dorsal stripe, yellow = ventral stripe, green = yolk sac stripe, cyan = lateral stripes, white = RPE, gray = other body melanin, purple = lens. As compared to wild-type larvae, *slc24a5*[b1/b1] mutant larvae exhibit a reduction in staining throughout the body and eyes (**D**) but retain the overall organization of pigment layers (**E**). The RPE in *slc24a5*[b1/b1] larvae (**F**) is thinner than the wild-type RPE (**C**) with less intense staining. The *mitfa*[w2/w2] mutant lacks all body pigment; staining is observed only in the RPE and some argentaffin material posterior to the eyes (**G–H**, arrow). The RPE of the *mitfa*[w2/w2] mutant (**I**) is of similar thickness and staining intensity as the wild-type RPE (**C**). For each comparison, visualization settings were kept constant for the wild-type and mutant fish. § = head segment only shown. All stained samples analyzed in this study are shown in *Figure 4—figure supplement 1*.

The online version of this article includes the following figure supplement(s) for figure 4:

**Figure supplement 1.** Overview of all analyzed samples.

thickness and intensity to the wild type (*Figure 4I*). All wild-type and mutant larvae analyzed for this study are shown in *Figure 4—figure supplement 1*.

To quantify the differences observed between the wild-type and mutant larvae, we prepared and micro-CT imaged *slc24a5*[b1/b1] (n = 3) and *mitfa*[w2/w2] (n = 3) 5 dpf larvae and segmented them into pigment regions as described above for computational analysis of both melanin volume and content. Because the *mitfa*[w2/w2] fish did not exhibit body pigment, only the head segments were used for the statistical analysis. For each pigment region, a statistically significant difference between the groups was found using a series of one-way ANOVAs (*Table 2*). While the melanin stripes were compared only between the wild-type and *golden* samples, total melanin, RPE melanin, and other melanin were

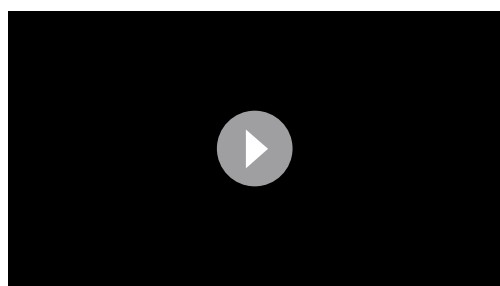

**Video 2.** Wild-type vs. mutant melanin patterns. This video shows a comparison between a representative wild-type 5 days post-fertilization (dpf) zebrafish larva and two representative pigmentation mutant 5 dpf larvae which have been stained with silver and X-ray microtomography (micro-CT) imaged. Visualization settings have been kept constant to facilitate direct comparisons between normalized samples. In volume renderings, the *golden* mutant (*slc24a5*$^{b1/b1}$) shows markedly reduced melanin content throughout the body and eyes. The *nacre* mutant (*mitfa*$^{w2/w2}$, head scan rendering only) shows essentially no body pigment but near-normal retinal pigment epithelia (RPE). Finally, stained volumes segmented by anatomic regions show the *golden* mutant exhibiting wild-type overall pigment organization while the *nacre* mutant (head scan rendering only) lacks the body stripes. Background music by DJ Vanselow.

https://elifesciences.org/articles/68920/figures#video2

compared between all three genotypes. For these regions, Tukey post hoc tests were used to determine which groups differed significantly from wild type (*Tables 3* and *4*).

Overall, the *slc24a5*$^{b1/b1}$ larvae exhibited a 75.3 % decrease in mean segmented melanin volume (p = 0.000005) from wild-type while the *mitfa*$^{w2/w2}$ fish exhibited a 63.1 % decrease (p = 0.000015, *Figure 5A*). Total melanin content (the cumulative sum of intensity in the segmented regions) followed a similar trend with an 83.8 % decrease in *slc24a5*$^{b1/b1}$ larvae (p = 0.000004) and a 55.0 % decrease in *mitfa*$^{w2/w2}$ larvae (p = 0.000040, *Figure 5B*). We also compared the melanin volume and content for the combined right and left RPE (*Figure 5C–D*), which contained the majority of the pigment for all samples, and the non-eye body melanin (*Figure 5E–F*). The *slc24a5*$^{b1/b1}$ RPE exhibited a 65.4 % decrease in melanin volume (p = 0.001) and an 80.5 % decrease in melanin content from wild-type RPE (p = 0.000058). Although the *mitfa*$^{w2/w2}$ mutant is generally considered to have normal eye pigmentation, we found that it exhibited a 28.5 % decrease in melanin content from the wild-type RPE (p = 0.014), potentially linked to an observed 25.1 % decrease in RPE volume, which was not statistically significant (p = 0.06). Interestingly, because RPE melanin made up 99.9 % (SD = ± 0.02%) of the observed staining in the *mitfa*$^{w2/w2}$

larvae, this smaller decrease in eye melanin content still left the *mitfa*$^{w2/w2}$ fish with more total melanin than the *golden* fish, which had consistently and dramatically diminished melanin throughout the body and eyes (*Figure 5B, D, F*).

## Conclusions and future work

Vertebrate models such as zebrafish play a key role in understanding human health and disease by elucidating relationships between mutations or environmental exposures and observable phenotypic changes, including pigmentation-related disorders such as albinism and melanoma. Ideally, the quantitative study of individual cell types in the 3D context of the whole organism would inform such relationships. In this study, we investigated whole-animal melanin pigmentation in wild-type and mutant zebrafish through the novel application of ionic silver staining for micro-CT imaging. Using this method, we were able to not only faithfully reconstruct pigmentation patterns of the larval zebrafish but moreover segment these patterns into anatomical regions and calculate relative pigmentation statistics within and between samples. Silver staining for melanin micro-CT has applications for other model organisms, as well as human biopsies for diseases like melanoma, and represents a model for the 3D quantitative analysis of specific biological features using micro-CT.

Micro-CT imaging of silver-stained samples provides a number of benefits for the analysis of melanin. The silver staining technique described here appears to be highly specific for melanin. Micro-CT provides high-resolution 3D images which are inherently digital reconstructions of the scanned sample, directly enabling quantification of signal intensity (melanin content). A wide FOV and 3D registration of multiple scans allowed us to investigate the entire volumes of our 5 dpf wild-type and mutant larvae. Preliminary experiments have indicated that the technique is equally applicable to larger samples, which are more difficult to investigate with light-based microscopy techniques but are readily accessible using micro-CT.

**Table 2.** Statistical analysis of wild-type, *slc24a5*[b1/b1], and *mitfa*[w2/w2] samples for segmented pigment regions.

| Pigment region | Segmented volume statistics | |
| --- | --- | --- |
| | One-way ANOVA F-statistic | p-Value |
| Total | F(2,6) = 178.43 | 0.000005 |
| RPE (right) | F(2,6) = 36.37 | 0.000442 |
| RPE (left) | F(2,6) = 21.77 | 0.002 |
| RPE (total) | F(2,6) = 29.64 | 0.001 |
| Dorsal stripe | F(1,4) = 145.98 | 0.000269 |
| Ventral stripe | F(1,4) = 376.02 | 0.000042 |
| Yolk sac stripe | F(1,4) = 22.04 | 0.009 |
| Lateral stripe (right) | F(1,4) = 39.01 | 0.003 |
| Lateral stripe (left) | F(1,4) = 17.48 | 0.014 |
| Other | F(2,6) = 52.86 | 0.000155 |
| Body (total) | F(2,6) = 939.89 | 0.000000032209 |

| Pigment region | Cumulative sum of intensity statistics | |
| --- | --- | --- |
| | One-way ANOVA F-statistic | p-Value |
| Total | F(2,6) = 185.76 | 0.000004 |
| RPE (right) | F(2,6) = 54.69 | 0.000141 |
| RPE (left) | F(2,6) = 70.40 | 0.000068 |
| RPE (total) | F(2,6) = 70.50 | 0.000068 |
| Dorsal stripe | F(1,4) = 138.02 | 0.000300 |
| Ventral stripe | F(1,4) = 1770.94 | 0.000002 |
| Yolk sac stripe | F(1,4) = 35.30 | 0.004 |
| Lateral stripe (right) | F(1,4) = 56.79 | 0.002 |
| Lateral stripe (left) | F(1,4) = 17.62 | 0.014 |
| Other | F(2,6) = 166.28 | 0.000006 |
| Body (total) | F(2,6) = 3690.79 | 0.00000000049447 |

F-statistics for each comparison are reported as F(degrees of freedom between groups, degrees of freedom within groups)= F-statistic. p < 0.05 indicates a significant difference between groups. While the melanin stripes were compared between the wild-type and *golden* samples, total melanin, RPE melanin, and other melanin were compared between all three genotypes. For these regions, Tukey post hoc tests were used to determine which groups differed significantly from wild type.

Micro-CT imaging involves fixation and staining of samples and exposure to high flux X-rays. Therefore, future studies of melanin at different stages of development or during experimental time courses using this method will require multiple samples of different ages with enough individuals of each stage to account for normal variation in zebrafish pigmentation. It is also worth noting that the quantitative measurements described here are relative to each other. For our measurements, we assumed that silver deposition was stoichiometric with the local concentration of melanin such that the arbitrary units of reconstructed intensity were proportional to the amount of stained material in the original sample. By performing our staining under the same conditions and for the same amount of time for all of our samples, and normalizing our reconstructions to a common background material, we were able to compare our samples directly. Because of these potential variables, staining conditions should be controlled for each experiment and measurements interpreted solely within the context of individual experiments.

We anticipate many future applications for the quantitative measurement of melanin in its native 3D context, both for zebrafish and for other samples of similar scale. Silver staining and micro-CT imaging may more thoroughly describe zebrafish mutants that have already been identified with

**Table 3.** Quantification of volume and cumulative sum of intensity for $slc24a5^{b1/b1}$ samples for segmented pigment regions.

| $slc24a5^{b1/b1}$ specimen | #1 | #2 | #3 | Average | SD | % Change from wt | p-Value |
|---|---|---|---|---|---|---|---|
| **Pigment region** | **Segmented volume (μm³)** | | | | | | |
| Total | $1.14 \times 10^6$ | $1.32 \times 10^6$ | $1.14 \times 10^6$ | $1.20 \times 10^6$ | $\pm 1.06 \times 10^5$ | −75.3 | 0.000005* |
| RPE (right) | $3.45 \times 10^5$ | $4.60 \times 10^5$ | $4.49 \times 10^5$ | $4.18 \times 10^5$ | $\pm 6.38 \times 10^4$ | −64.2 | 0.000362* |
| RPE (left) | $4.08 \times 10^5$ | $4.16 \times 10^5$ | $4.05 \times 10^5$ | $4.10 \times 10^5$ | $\pm 5.45 \times 10^3$ | −66.6 | 0.002* |
| Dorsal stripe | $9.70 \times 10^4$ | $1.74 \times 10^5$ | $8.36 \times 10^4$ | $1.18 \times 10^5$ | $\pm 4.87 \times 10^4$ | −77.0 | 0.000269** |
| Ventral stripe | $2.33 \times 10^5$ | $2.06 \times 10^5$ | $1.44 \times 10^5$ | $1.94 \times 10^5$ | $\pm 4.56 \times 10^4$ | −87.3 | 0.000042** |
| Yolk sac stripe | $3.97 \times 10^4$ | $3.93 \times 10^4$ | $2.07 \times 10^4$ | $3.32 \times 10^4$ | $\pm 1.09 \times 10^4$ | −87.5 | 0.009** |
| Lateral stripe (right) | $4.80 \times 10^3$ | $7.58 \times 10^3$ | $4.53 \times 10^3$ | $5.63 \times 10^3$ | $\pm 1.69 \times 10^3$ | −89.5 | 0.003** |
| Lateral stripe (left) | $7.16 \times 10^3$ | $9.09 \times 10^3$ | $3.99 \times 10^3$ | $6.75 \times 10^3$ | $\pm 2.58 \times 10^3$ | −86.5 | 0.014** |
| Other | $6.83 \times 10^3$ | $1.32 \times 10^4$ | $3.01 \times 10^4$ | $1.67 \times 10^4$ | $\pm 1.20 \times 10^4$ | −74.2 | 0.001* |
| **Pigment region** | **Cumulative sum of intensity (normalized units)** | | | | | | |
| Total | $7.54 \times 10^{10}$ | $8.51 \times 10^{10}$ | $7.50 \times 10^{10}$ | $7.85 \times 10^{10}$ | $\pm 5.74 \times 10^9$ | −83.7 | 0.000004* |
| RPE (right) | $2.43 \times 10^{10}$ | $3.17 \times 10^{10}$ | $3.11 \times 10^{10}$ | $2.90 \times 10^{10}$ | $\pm 4.07 \times 10^9$ | −79.8 | 0.000125* |
| RPE (left) | $2.98 \times 10^{10}$ | $3.17 \times 10^{10}$ | $2.84 \times 10^{10}$ | $3.00 \times 10^{10}$ | $\pm 1.69 \times 10^9$ | −81.2 | 0.000056* |
| Dorsal stripe | $5.89 \times 10^9$ | $8.87 \times 10^9$ | $5.28 \times 10^9$ | $6.68 \times 10^9$ | $\pm 1.92 \times 10^9$ | −84.5 | 0.000300** |
| Ventral stripe | $1.16 \times 10^{10}$ | $9.44 \times 10^9$ | $7.10 \times 10^9$ | $9.39 \times 10^9$ | $\pm 2.26 \times 10^9$ | −91.0 | 0.000002** |
| Yolk sac stripe | $2.59 \times 10^9$ | $1.84 \times 10^9$ | $1.26 \times 10^9$ | $1.90 \times 10^9$ | $\pm 6.66 \times 10^8$ | −89.5 | 0.004** |
| Lateral stripe (right) | $3.11 \times 10^8$ | $4.07 \times 10^8$ | $2.75 \times 10^8$ | $3.31 \times 10^8$ | $\pm 6.85 \times 10^7$ | −93.1 | 0.002** |
| Lateral stripe (left) | $4.81 \times 10^8$ | $4.52 \times 10^8$ | $2.44 \times 10^8$ | $3.92 \times 10^8$ | $\pm 1.30 \times 10^8$ | −90.2 | 0.014** |
| Other | $4.21 \times 10^8$ | $7.36 \times 10^8$ | $1.37 \times 10^9$ | $8.43 \times 10^8$ | $\pm 4.84 \times 10^8$ | −84.0 | 0.000015* |

*p-Values determined by Tukey post hoc test following one-way ANOVA. **p-Values determined directly by one-way ANOVA. p-Values were considered significant <0.05. Raw data provided in Table 3—source data 1.

The online version of this article includes the following source data for table 3:

**Source data 1.** Source data for quantification of volume and cumulative sum of intensity for $slc24a5^{b1/b1}$ samples for segmented pigment regions.

effects on larval, juvenile, and adult pigmentation (*Parichy, 2006*), and our work with the *nacre/casper* zebrafish highlights the potential to identify new mutants with subtler effects on pigment localization or concentration. Zebrafish are frequently used for studying complex animal pigmentation patterning, including computational models of pigment cell interactions (*Owen et al., 2020*) or studies of human skin color (*Lamason et al., 2005*). High-resolution, whole-animal 3D images of melanin could help experimentally confirm such studies.

In addition to its use in the study of pigmentation, zebrafish melanin is also studied in the context of models of melanoma. Darkly pigmented lesions in transgenic or transplanted zebrafish can be tracked and scored to investigate the genetic underpinnings of melanoma origin and spread (*Ceol et al., 2011*; *Fazio et al., 2021*); silver staining and micro-CT imaging could enable computational phenotyping and direct detailed morphological analysis of lesions in their native context. Melanin is also a common pigment in many mammals, including humans (*Slominski et al., 2004*). Human mm-to cm-scale melanocytic nevi or melanoma biopsy samples may be stained for micro-CT imaging as described in this study to enable quantitative morphological analyses of these samples. Silver staining may also be combined with other histotomography stains, such as phosphotungstic acid, to provide tissue context for melanin localization. Dual-energy CT may be used to deconvolute such multi-stained samples into component elemental materials (*Liu et al., 2009*).

While the work presented here is an example of quantitative analysis of a particular cellular target, it also represents a step toward the ideal of quantitative morphological analysis of any cell type in

**Table 4.** Quantification of volume and cumulative sum of intensity for *mitfa*[w2/w2] samples for segmented pigment regions.

| *mitfa*[w2/w2] specimen | #1[§] | #2[§] | #3[§] | Average | SD | % Change from wt | p-Value |
|---|---|---|---|---|---|---|---|
| **Pigment region** | | | | **Segmented volume (µm³)** | | | |
| Total | $1.64 \times 10^6$ | $2.03 \times 10^6$ | $1.72 \times 10^6$ | $1.80 \times 10^6$ | $\pm 2.04 \times 10^5$ | −63.1 | 0.000015* |
| RPE (right) | $7.99 \times 10^5$ | $1.00 \times 10^6$ | $7.93 \times 10^5$ | $8.64 \times 10^5$ | $\pm 1.18 \times 10^5$ | −26.0 | 0.032* |
| RPE (left) | $8.41 \times 10^5$ | $1.02 \times 10^6$ | $9.21 \times 10^5$ | $9.28 \times 10^5$ | $\pm 9.09 \times 10^4$ | −24.2 | 0.120* |
| Other | $4.22 \times 10^3$ | $5.02 \times 10^3$ | $5.29 \times 10^3$ | $4.84 \times 10^3$ | $\pm 5.57 \times 10^2$ | −92.5 | 0.000169* |
| **Pigment region** | | | | **Cumulative sum of intensity (normalized units)** | | | |
| Total | $1.97 \times 10^{11}$ | $2.64 \times 10^{11}$ | $1.91 \times 10^{11}$ | $2.17 \times 10^{11}$ | $\pm 4.05 \times 10^{10}$ | −55.0 | 0.000040* |
| RPE (right) | $1.01 \times 10^{11}$ | $1.33 \times 10^{11}$ | $8.80 \times 10^{10}$ | $1.07 \times 10^{11}$ | $\pm 2.30 \times 10^{10}$ | −25.4 | 0.040* |
| RPE (left) | $9.54 \times 10^{10}$ | $1.31 \times 10^{11}$ | $1.03 \times 10^{11}$ | $1.10 \times 10^{11}$ | $\pm 1.86 \times 10^{10}$ | −31.2 | 0.010* |
| Other | $2.40 \times 10^8$ | $2.70 \times 10^8$ | $2.77 \times 10^8$ | $2.62 \times 10^8$ | $\pm 1.98 \times 10^7$ | −95.0 | 0.000007* |

*p-Values determined by Tukey post hoc test following one-way ANOVA. § = head segment only analyzed. p-Values were considered significant <0.05. Raw data provided in Table 4—source data 1.

The online version of this article includes the following source data for table 4:

**Source data 1.** Source data for quantification of volume and cumulative sum of intensity for *mitfa*[w2/w2] samples for segmented pigment regions.

the context of a whole organism. Ideal morphological phenotyping is volumetric, because biological structures are 3D, and quantitative, to extend analysis from simple observation to automation, modeling, and prediction. Micro-CT images are inherently both quantitative and 3D, enabling computational analysis of large samples that may be opaque to optical microscopy techniques. The development of new staining protocols for micro-CT will expand the capabilities of this imaging modality, particularly for specific molecular or cellular targets. Notably, silver-based staining has been used previously for micro-CT imaging using antibody-mediated deposition for molecular targets in chick embryos (*Metscher and Müller, 2011*); our work suggests that melanin-mediated silver deposition may confound such a strategy in pigmented samples. However, silver staining of melanin also opens the possibility of targeted melanin deposition for micro-CT molecular imaging using, for example, the prophenoloxidase reporter system in *Drosophila* (*True et al., 2001*). The development of analytical pipelines with the potential for machine learning and automation is equally critical since manual segmentation projects like the one reported here will soon be overwhelmed by the amount of data generated by high-throughput, high-dimensional imaging.

One frequent criticism of micro-CT is the concern of access, both to imaging technology and to micro-CT data. Indeed, for this proof-of-concept study of silver staining, competitive synchrotron beamtime necessitated prioritization of limited sample sizes. Improved imaging resources, such as laboratory micro-CT setups that provide synchrotron-like imaging capabilities, and beamlines at established synchrotrons dedicated to wide-field, high-resolution micro-CT, would increase access and throughput. As more and larger samples are scanned through these resources, data storage and sharing bottlenecks will need to be addressed. The raw micro-CT reconstructions described in this study were on the order of 80–100 GB per scan, which can be challenging to work with for users with only standard computing resources or without familiarity with 3D image analysis. To make our 3D pigmentation renderings more generally accessible without requiring the user to download full volumes, we have expanded our online, open-source ViewTool (http://3D.fish) (*Ding et al., 2019*) to include a browser-based volumetric viewer (http://3D.fish/stains/silver) based on the open-source Three.js project (https://threejs.org). This volumetric viewer contains all of the wild-type and mutant samples described in this study, with customizable visualization options and toggles for viewing the segmented pigment region labels.

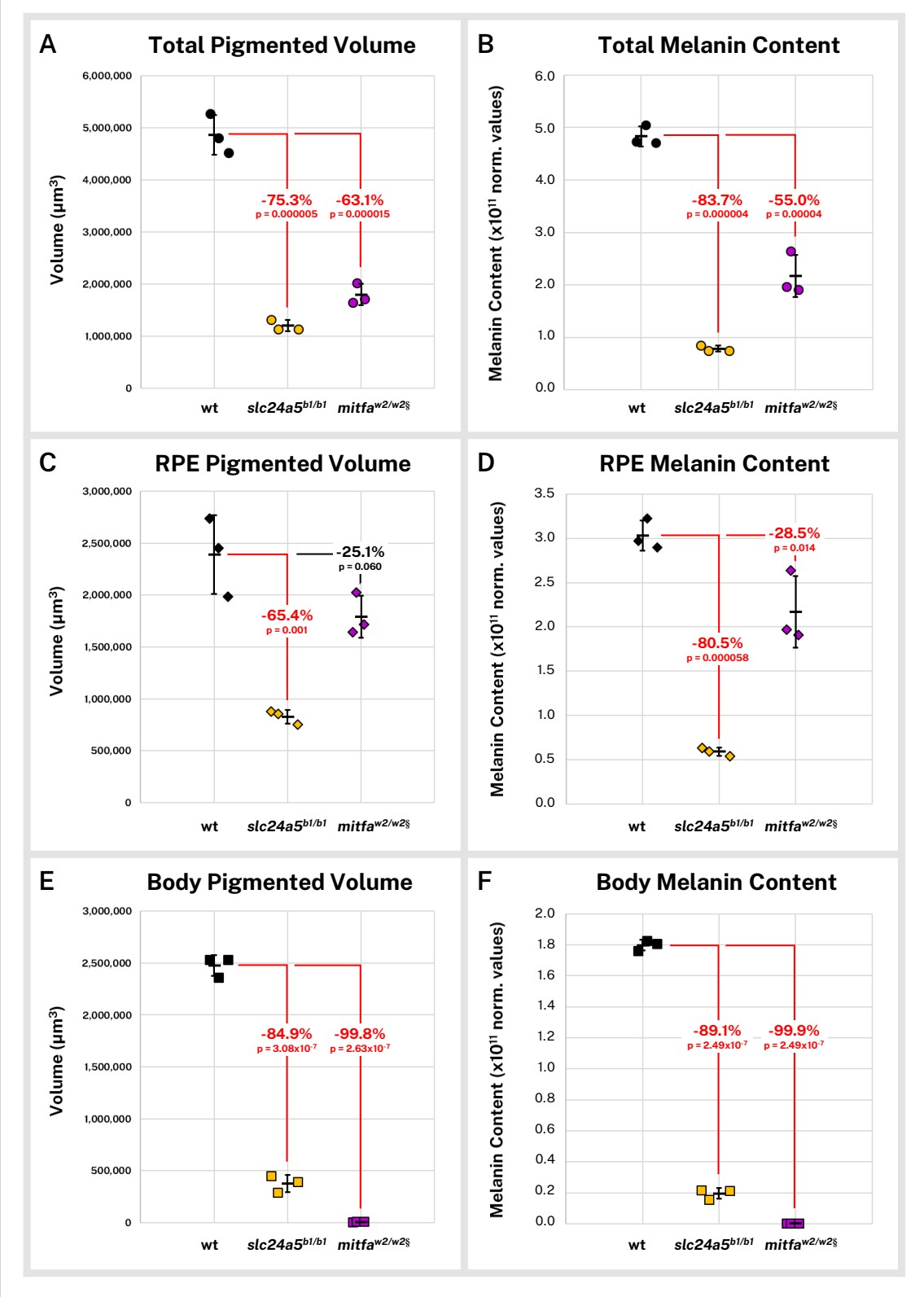

**Figure 5.** Silver-based X-ray microtomography (micro-CT) enables quantitative comparisons of wild-type and mutant pigmented samples. Wild-type (wt, n = 3), *slc24a5*[b1/b1] (n = 3), and *mitfa*[w2/w2] (n = 3) 5 days post-fertilization (dpf) larvae were stained with silver and micro-CT imaged under the same conditions. Normalized reconstructed intensity values (norm. values) are assumed proportional to melanin density; integrated intensity values for the segmented regions represent the melanin content of these regions. Volume and melanin content from total pigmented regions (**A–B**), combined right

*Figure 5 continued on next page*

*Figure 5 continued*

and left retinal pigment epithelia (RPE) (**C–D**), and pigmented regions outside the eye (**E–F**) are shown for all samples with percent change of mean from wt indicated. p-Values were determined by Tukey post hoc test following one-way ANOVA and considered significant at p < 0.05 (shown in red). § = head segments only analyzed. Error bars = standard deviation from average.

In summary, we used silver deposition micro-CT imaging to quantitatively characterize normal and mutant zebrafish melanin distribution and content in the 3D context of whole larvae. The method described herein may be applied to other samples of similar scale, including additional zebrafish models of pigment development or disease and human biopsy samples, and has implications for the future development of micro-CT staining for computational phenomics.

## Acknowledgements

We thank Dr Patrick La Rivière for helpful discussion about micro-CT analysis, and Margaret Hubley and Kathryn Early for assistance raising and preparing zebrafish for this work. Funding: This research was supported by the National Institutes of Health (PI: KCC, R24-RR017441, and PI: KCC, R24-OD018559), the Jake Gittlen Laboratories for Cancer Research, the Penn State College of Medicine Zebrafish Functional Genomics Core, the Huck Institutes of the Life Sciences and the Institute for Cyber Science, Penn State University, and the Pennsylvania Department of Health Tobacco CURE Funds. The Pennsylvania Department of Health specifically disclaims responsibility for any analysis, interpretations, or conclusions. This research used resources of the ALS, a U.S. DOE Office of Science User Facility under contract no. DE-AC02-05CH11231.

## Additional information

### Competing interests

Yuxin Wang: Yuxin Wang is affiliated with Mobile Imaging Innovations, Inc. The author has no other competing interests to declare.. The other authors declare that no competing interests exist.

### Funding

| Funder | Grant reference number | Author |
|---|---|---|
| NIH Office of the Director | R24-OD018559 | Keith C Cheng |
| National Institutes of Health | R24-RR017441 | Keith C Cheng |
| Penn State College of Medicine | Jake Gittlen Laboratories for Cancer Research | Keith C Cheng |
| Penn State College of Medicine | Zebrafish Functional Genomics Core | Keith C Cheng |
| Huck Institutes of the Life Sciences | | Keith C Cheng |
| Pennsylvania State University | Institute for Cyber Science | Keith C Cheng |
| Pennsylvania Department of Health | Tobacco CURE Funds | Keith C Cheng |

The funders had no role in study design, data collection and interpretation, or the decision to submit the work for publication.

### Author contributions

Spencer R Katz, Conceptualization, Data curation, Formal analysis, Investigation, Methodology, Resources, Validation, Visualization, Writing – original draft, Writing – review and editing; Maksim A Yakovlev, Data curation, Formal analysis, Investigation, Methodology, Resources, Software; Daniel J Vanselow, Data curation, Formal analysis, Investigation, Methodology, Resources, Software,

Supervision, Visualization, Writing – review and editing; Yifu Ding, Alex Y Lin, Methodology, Resources, Writing – review and editing; Dilworth Y Parkinson, Methodology, Project Administration: LBNL ALS, Project administration, Resources, Supervision; Yuxin Wang, Methodology, Resources, Supervision; Victor A Canfield, Supervision, Validation, Writing – review and editing; Khai C Ang, Investigation, Project administration, Resources, Supervision, Validation, Writing – review and editing; Keith C Cheng, Conceptualization, Funding acquisition, Methodology, Project administration, Resources, Supervision, Validation, Writing – original draft, Writing – review and editing

### Author ORCIDs
Spencer R Katz http://orcid.org/0000-0002-5586-3562
Maksim A Yakovlev http://orcid.org/0000-0003-1846-3751
Daniel J Vanselow http://orcid.org/0000-0002-9221-8634
Yifu Ding http://orcid.org/0000-0002-4629-5858
Alex Y Lin http://orcid.org/0000-0002-1653-4168
Victor A Canfield http://orcid.org/0000-0002-4359-1790
Khai C Ang http://orcid.org/0000-0001-7695-9953
Keith C Cheng http://orcid.org/0000-0002-5350-5825

### Ethics
All procedures on live animals were approved by the Institutional Animal Care and Use Committee (IACUC) at the Pennsylvania State University College of Medicine, ID: PROTO201800300, "Developing Tissue-, Cell-, and Protein-specific Staining for histo-tomography, a form of X-ray Microtomography (microCT)."

### Decision letter and Author response
Decision letter https://doi.org/10.7554/eLife.68920.sa1
Author response https://doi.org/10.7554/eLife.68920.sa2

## Additional files

### Supplementary files
• Transparent reporting form

### Data availability
Source Data files have been provided with the manuscript for each table and the measurements in Figure 2-figure supplement 1. Reconstructions, processed normalized reconstructions, source code, and Avizo project and source files necessary for reproducing the analyses in the manuscript are publicly available in the Dryad Digital Repository (https://doi.org/10.5061/dryad.wwpzgmsjn). To make our data more readily available for users without extensive imaging backgrounds or computing resources, our open-source online 3D viewer, ViewTool, is publicly available at http://3D.fish/stains/silver.

The following dataset was generated:

| Author(s) | Year | Dataset title | Dataset URL | Database and Identifier |
| --- | --- | --- | --- | --- |
| Katz SR, Yakovlev MA, Vanselow DJ, Ding Y, Lin AY, Parkinson DY, Wang Y, Canfield VA, Ang KC, Cheng KC | 2021 | Data from: Whole-organism 3D quantitative characterization of zebrafish melanin by silver deposition micro-CT | https://doi.org/10.5061/dryad.wwpzgmsjn | Dryad Digital Repository, 10.5061/dryad.wwpzgmsjn |

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
