## [Decision Letter]

**Acceptance summary:**

In this article, Katz, Cheng and colleagues apply microCT to detect melanin in zebrafish. This is a very natural extension of the groundbreaking work they have done previously to use microCT as a means of "deep phenotyping" of zebrafish mutants. This type of phenotyping is important since conventional imaging likely misses important phenotypes that are otherwise important. Applying microCT to melanin is useful, since disorders of pigmentation are very common in zebrafish and the genes underlying this have been shown to have direct relevance to human conditions such as vitiligo and melanoma. They cleverly use silver staining, a common method in histology, to a whole animal, to detect patterns of melanin. This work will be immediately useful to those interested not only in pigmentation biology, but opens up avenues for other types of complex phenotyping using microCT.

**Decision letter after peer review:**

Thank you for submitting your article "Whole-organism 3D quantitative characterization of zebrafish melanin by silver deposition micro-CT" for consideration by *eLife*. Your article has been reviewed by 2 peer reviewers, including Richard White as the Senior and Reviewing Editor and Reviewer #3. The following individual involved in review of your submission has agreed to reveal their identity: Stephen C Ekker (Reviewer #2).

The reviewers have discussed their reviews with one another, and the Editor has drafted this to help you prepare a revised submission.

Essential revisions:

In this manuscript, Katz and colleagues apply their microCT method to detection of melanin. This is a very natural followup to their previous paper describing the use of microCT for whole animal phenotyping in zebrafish. What is clever about this new study is that they adopted the use of silver stain, a common way to stain melanin in histology slides, to whole animal use. Because silver has so reliably been used for decades, this gave them a high chance of it working in whole animals once they worked out the staining techniques. They show that this silver staining appropriately marks zebrafish stripes and eyes (the biggest melanin producing organs in the body) and that they could detect differences in the golden, nacre, and casper mutants. Overall, the results are convincing and exciting. While it is clear that microCT will remain, at least for the time being, a fairly specialized technique, the ability to apply it more broadly will be an exciting area for future development as both the hardware and software improve. Only a few items that they could address to make the study stronger. None of these are required or require new experiments, but only suggestions if you have the data:

1) Detection of melanin in internal organs

Although the RPE and skin stripes are the most obvious melanin containing organs, two other structures are known to have melanin: the substantia nigra (as you point out in the introduction) as well as the kidney (Kapp, et al., Nature. 2018 Jun; 558(7710): 445-448.). The kidney is especially interesting since melanin there is thought to have protective roles, and kidney melanin is absent in the casper mutants. In the images you have already taken, it would be useful to see if you could quantify the substantia nigra and kidney in the WT versus mutant lines (golden, casper, etc). This would lend further credence to the idea that microCT really offers advantages over conventional light or fluorescent microscopy, since you could image internal organs. We realize this may not be possible, and I do not consider this essential for publication, but any data using the images you already have would add to the manuscript.

2) Other ages

You mention future applications to different stages, but in reality it would be useful to know how you could apply this method to both younger fish (i.e. 24hpf) or much older fish (i.e. adults). Have you tried any imaging of melanin at any stages other than 5dpf. Again, I do not consider this essential, but would be helpful if you have such data.

3) Line 339-340, you state "To examine these mutants, we bred mitfaw2/w2; mpv17a9/+ larvae to produce a mix of both nacre and casper offspring, which are indistinguishable from each other at 5 dpf", but this is technically not true. From a melanocyte perspective they are indistinguishable , but they have a very clear difference in terms of the iridophore phenotype (casper is missing these, nacre is not). I would clarify this. This difference was noted in the original casper manuscript (White et al., Cell Stem Cell 2008) and this reference should be added to the paper. Throughout the manuscript please make sure you distinguish between nacre (just mitfa mutant) versus casper (mitfa/mpv17 mutant).

4) Why do you need microCT to be this quantitative? Why cannot optical imaging with color quantitation be used instead to solve this technical question (methods that are far more accessible than microCT)?

5) What are the throughput constraints? There are not very many individual quantitative data points, raising a question of longitudinal capacity.

6) Figure 1 D – right; why do we see body but no eye pigment silver staining?

---

## [Author Response]

Essential revisions:In this manuscript, Katz and colleagues apply their microCT method to detection of melanin. This is a very natural followup to their previous paper describing the use of microCT for whole animal phenotyping in zebrafish. What is clever about this new study is that they adopted the use of silver stain, a common way to stain melanin in histology slides, to whole animal use. Because silver has so reliably been used for decades, this gave them a high chance of it working in whole animals once they worked out the staining techniques. They show that this silver staining appropriately marks zebrafish stripes and eyes (the biggest melanin producing organs in the body) and that they could detect differences in the golden, nacre, and casper mutants. Overall, the results are convincing and exciting. While it is clear that microCT will remain, at least for the time being, a fairly specialized technique, the ability to apply it more broadly will be an exciting area for future development as both the hardware and software improve. Only a few items that they could address to make the study stronger. None of these are required or require new experiments, but only suggestions if you have the data:1) Detection of melanin in internal organsAlthough the RPE and skin stripes are the most obvious melanin containing organs, two other structures are known to have melanin: the substantia nigra (as you point out in the introduction) as well as the kidney (Kapp, et al., Nature. 2018 Jun; 558(7710): 445-448.). The kidney is especially interesting since melanin there is thought to have protective roles, and kidney melanin is absent in the casper mutants. In the images you have already taken, it would be useful to see if you could quantify the substantia nigra and kidney in the WT versus mutant lines (golden, casper, etc). This would lend further credence to the idea that microCT really offers advantages over conventional light or fluorescent microscopy, since you could image internal organs. We realize this may not be possible, and I do not consider this essential for publication, but any data using the images you already have would add to the manuscript.

We agree that a major advantage of micro-CT over conventional microscopy is the ability to image and quantify internal structures, including, in this case, internal melanin. We also share the goal of imaging melanin-containing organs besides the larval zebrafish stripes and eyes in future samples, including those in other organisms. For example, while zebrafish do have dopaminergic neurons in the adult hindbrain, there are far fewer cells than in the mammalian midbrain substantia nigra, and they do not appear to be associated with concentrated neuromelanin pigmentation. While outside the scope of the present paper, biopsies from substantia nigra of mice or rats would make excellent samples for testing the broader applicability of our silver staining protocol for micro-CT.

By comparing the larval figures in the Kapp, et al., 2018, paper with our silver-stained micro-CT images, we believe that the umbrella of protective melanin covering the pronephric kidney belongs to the ventral stripe, which continues on above the yolk and down toward the tail. This stripe is indeed absent in the nacre/casper samples. The Kapp paper also mentions that the adult zebrafish kidneys are “covered” with melanophores. While identifying kidney-associated melanin will be included in future imaging of older zebrafish samples (see below), we would like to point out that KCC has observed in unpublished work under a dissecting microscope that melanophores indeed exist around the kidney, but their appearance did not suggest any intimate relationship with tubules or glomeruli in the kidney parenchyma, as is the case around photoreceptors of the retina.

2) Other agesYou mention future applications to different stages, but in reality it would be useful to know how you could apply this method to both younger fish (i.e. 24hpf) or much older fish (i.e. adults). Have you tried any imaging of melanin at any stages other than 5dpf. Again, I do not consider this essential, but would be helpful if you have such data.

Expanding the applications of this staining technique to other stages of zebrafish development, and to other organisms, is an active area of research for our lab. In the initial development of the silver staining, we tested our technique on 3dpf zebrafish with similar efficacy as to the 5dpf fish, lending credence to the idea that zebrafish of similar size/stage (e.g., larval fish) can be handled essentially the same way for micro-CT preparation. We are especially interested in optimizing silver staining for whole adult samples, which can fit in our expanded field-of-view custom detector. Adult zebrafish tissue is optically opaque and therefore a poor candidate for traditional microscopic analysis. As of this writing, we have performed trial silver staining on a 96dpf wild-type adult zebrafish with micro-CT imaging and reconstruction of a single segment of the head. While initial reconstructions seem favorable for melanin staining, further work is needed to optimize silver staining, provide proper controls, image whole individuals, and perform quantitative analyses of adult melanin. Since the synchrotrons needed to image our samples have yet to reopen since COVID shutdown, we have not had opportunity to pursue these studies. It is therefore more appropriate to share/publish the adult data in a more complete form, in a subsequent manuscript.

3) Line 339-340, you state "To examine these mutants, we bred mitfaw2/w2; mpv17a9/+ larvae to produce a mix of both nacre and casper offspring, which are indistinguishable from each other at 5 dpf", but this is technically not true. From a melanocyte perspective they are indistinguishable , but they have a very clear difference in terms of the iridophore phenotype (casper is missing these, nacre is not). I would clarify this. This difference was noted in the original casper manuscript (White, et al., Cell Stem Cell 2008) and this reference should be added to the paper. Throughout the manuscript please make sure you distinguish between nacre (just mitfa mutant) versus casper (mitfa/mpv17 mutant).

Thank you for pointing out this missing reference. We have added it to the appropriate places in the manuscript. While we agree that *nacre* and *casper* larvae may be separated by the presence or absence of iridophores, the larvae used for this study were screened only by using melanophore phenotypes prior to staining, and we are unable to screen by iridophores after staining. Therefore, in the manuscript we refer to these fish by their mitfa^w2/w2^ background, or as *nacre/casper* fish. We have also updated the language both in the “Materials and methods” and the “Results” sections to clarify the references to these mutants throughout the manuscript.

4) Why do you need microCT to be this quantitative? Why cannot optical imaging with color quantitation be used instead to solve this technical question (methods that are far more accessible than microCT)?

One of the advantages of micro-CT imaging is that it is inherently quantitative, with digital data associated with a numerical value at each reconstructed voxel. This data is arranged in the three-dimensional context of the original sample. Quantitative measurements like the ones described here may eventually be used for modelling, machine learning, and automation that will enable comprehensive, high-dimensional phenotyping.

While we agree that color quantitation with optical imaging is currently more accessible for most labs, optical imaging is limited by melanin’s opacity to visible light, which is not the case for silver/melanin staining. While young zebrafish are generally considered transparent, optical imaging has limited ability to resolve deep tissue material, resulting in loss of resolution in deeper tissues (>~200 µm). Indeed, melanin itself is optically opaque and obscures underlying structures. As a result, surface imaging of zebrafish larvae misrepresent melanin content in overlapping melanophores or deep tissue melanin, and provides more limited information about the 3D architecture of the pigment. Older or larger samples in which thick soft tissue will scatter more light make optical imaging and quantitation even more challenging, while X-ray-based imaging like micro-CT can still provide full-thickness, quantitative imaging. Its value will admittedly require additional work from a variety of labs to realize its importance in answering questions we have yet to ask. Our work is but a first step.

We have added language to the “Conclusions and Future Work” section of the manuscript to emphasize this potential for quantitative micro-CT data.

5) What are the throughput constraints? There are not very many individual quantitative data points, raising a question of longitudinal capacity.

With competitive beamtime availability at national synchrotron resources, throughput for micro-CT imaging is currently a bottleneck. While this manuscript focused on proof-of-concept technique development for silver staining, for which a small number of diverse samples was prioritized, increasing imaging throughput is an area of active research for the lab. Current attempts to increase throughput include 1) creating dedicated synchrotron beamlines for biomedically-oriented micro-CT at beamlines like LBL (Lawrence Berkeley Laboratory) and ANL (Argonne National Laboratory), which a decade of work is beginning to yield results, 2) developing laboratory-based micro-CT technology with synchrotron-like resolution, 2) using pink beam (polychromatic X-rays) to decrease scan time, 3) multiplexing samples, 4) increasing field-of-view to cover more material per scan, 5) multi-resolution strategies whereby many samples can be screened at lower resolutions and select samples of interest that can subsequently be re-imaged and analyzed at higher resolutions.

We have expanded the “Conclusions and Future Work” section of the manuscript to acknowledge these limitations of access and throughput to micro-CT imaging, and discuss future considerations to realize its potential for larger-scale quantitative phenotyping. Indeed, this very publication will contribute towards greater awareness of what we think is the huge potential of derivatives of X-ray histotomography.

6) Figure 1 D – right; why do we see body but no eye pigment silver staining?

In this figure, we present 3D reconstructions of the silver staining in the wild-type fish, which includes both body and eye pigment. In Figure 1D (right), we show a top-down view of this reconstruction in a similar view to the light micrograph in Figure 1D (left); to make it easier to see the similar features in the dorsal melanin for both images, we added a red color to just the dorsal-most melanin in the micro-CT image. However, everything else in the image (in grayscale) still represents silver-stained material. We have updated the Figure 1 legend to clarify this view of the data.